# Detecting Erroneous Handwritten Byzantine Text Recognition

**John Pavlopoulos**
Department of Informatics
Athens University of Economics and Business
Greece
annis@aueb.gr

**Vasiliki Kougia**
Faculty of Computer Science
University of Vienna, Austria
UniVie Doctoral School Computer Science
vasiliki.kougia@univie.ac.at

**Paraskevi Platanou**
Department of Philology, School of Philosophy
University of Athens, Greece
pplatanou@phil.uoa.gr

**Holger Essler**
Department of Humanities
Ca'Foscari University of Venice, Italy
holger.essler@unive.it

## Abstract

Handwritten text recognition (HTR) yields textual output that comprises errors, which are considerably more compared to that of recognised printed (OCRed) text. Post-correcting methods can eliminate such errors but may also introduce errors. In this study, we investigate the issues arising from this reality in Byzantine Greek. We investigate the properties of the texts that lead post-correction systems to this adversarial behaviour and we experiment with text classification systems that learn to detect incorrect recognition output. A large masked language model, pre-trained in modern and fine-tuned in Byzantine Greek, achieves an Average Precision score of 95%. The score improves to 97% when using a model that is pre-trained in modern and then in ancient Greek, the two language forms Byzantine Greek combines elements from. A century-based analysis shows that the advantage of the classifier that is further-pre-trained in ancient Greek concerns texts of older centuries. The application of this classifier before a neural post-corrector on HTRed text reduced significantly the post-correction mistakes.

## 1 Introduction

Handwritten Text Recognition (HTR) concerns the automated transcription of a handwritten text, extracting its computerised form from an image. Applying HTR on an old manuscript can lead to a high character error rate (CER), often due to the lack of training data (Pavlopoulos et al., 2023). In the real world, this erroneous HTR output is post-corrected by experts, who get involved, typically, in a tedious and time-consuming task. Delaying the delivery of manually curated data, hinders both: the preservation of digitised manuscripts that have not yet been

Figure 1: Number of HTRed-lines according to their CER as presented by Pavlopoulos et al. (2023)

transcribed; and the progress of HTR for historical manuscripts, which struggles with low resources. This is why researchers from Digital Humanities have started encouraging the development of Natural Language Processing (NLP) solutions, in an attempt to assist the experts toward this goal, since improving the accuracy of recognition will be exciting for the disciplines struggling with vast corpora and little man power (McGillivray et al., 2020).

### 1.1 Background

Although detecting erroneous recognised output is not a novel task (Chiron et al., 2017; Amrhein and Clematide, 2018; Rigaud et al., 2019; Jatowt et al., 2019; Nguyen et al., 2020; Schaefer and Neudecker, 2020), existing published studies focus on printed material (i.e., OCRed). Handwritten text, on the other hand, increases the challenge in recognition compared to printed material and hence, leads to more errors. The different hands, the greater time period, morphology and spelling

| | Transcription | Recognition |
|---|---|---|
| a. | των ποδων | τωη ποδων |
| b. | των ποδων | ακοασδδεδ |
| a. | ἐγγινομένα πάθη μὴ σβεννύντε | ἐγγινομενα πάθη μὴ σβεννύντε |
| b. | ἐγγινομένα πάθη μὴ σβεννύντε | ἐγγινομένδπάθη μαε σβανησστε |
| a. | τες ἐμπυρίζουσι τὸν ἀμπελῶνα ἀλλὰ καὶ ὁ διὰ τες | τες ἐμπυρίζουσι τὸν ἀμπελονα ἀλλὰ καὶ ὁ δια τες |
| b. | τες ἐμπυρίζουσι τὸν ἀμπελῶνα ἀλλὰ καὶ ὁ διὰ τες | τεσἐμπυρίσοβσι κὸα ἀμπελοαα κἀλὰ κκκαὶ α διοτες |

Table 1: Synthetic transcription (human-generated) and recognition (system-generated) output with few (a) and many (b) errors. Finding the differences between the transcription and the recognition is easier for the three examples shown in red (b).

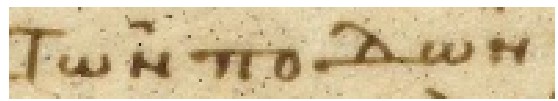

Figure 2: Image of the handwritten text 'τῶν ποδων' - in the recognised text, the third character was misspelled, written as eta ('η') instead of 'ν' (Table 1, 1st row)

(see Section 3), can explain the greater diversity of error rates (Fig. 1). The recognition quality, however, is related to the detection quality of erroneous output. To give an example, the detection of an erroneous recognition is easier when the recognition output is scrambled compared with a single incorrect character. This is shown in Table 1, where classifying recognised text (as erroneous/not) is easier for the human eye in (b) compared to that of (a). We hypothesise that this applies also to models.

We argue that *detecting erroneously recognised text can improve automatic error correction*. Current approaches to post-correcting the often flawed recognised text, disregard the likelihood of a flawless recognition, assuming that the correct text can be simply copied (e.g., with an encoder-decoder-based correction). Consequently, they are faced with the possibility of introducing, instead of mitigating, errors that would be absent if post-correction hadn't been applied. Taking the case of a misspelled article (e.g., see Fig. 2), for instance, there is a variety of letters to choose from, for replacement, always accordingly to the declension and the case of the noun. Even if grammatical rules are set (e.g., in a rule-based system), an

HTR system working at the character level is likely examining morphological aspects and often disregards semantic and pragmatic levels. A vowel replacement function addressing the morphological level, however, may omit important semantic aspects and relationships between sentence elements at the syntactic level and thus, misinterpret the case. For example, the line "λεντιον διεζωσεν εαυτόν" was HTRed as "λεντιον διεζωσενεαυτόν". A neural corrector, then, which received as input the HTRed output, yielded "λεντιον διεζωσεν εαυτον" (Pavlopoulos et al., 2023), correctly separating the two last words but incorrectly removing the accent. In this way and in this instance, post-correcting involves the risk of introducing errors by transforming probably correct output cases.

## 1.2 The focus of the study

This study quantifies the hazards of the above-mentioned adversarial post-correction, by focusing on the Byzantine Greek language. The development of this language entails a comprehensive examination of written language throughout the centuries under study. In essence, Byzantine Greek emerged as an academic style known as Atticised Greek, incorporating religious topics. As time progressed, elements of contemporary spoken Greek gradually became integrated into the language (Horrocks, 2014). As the results of our study also show, the combination of pre-trained data on Modern Greek, resembling what was considered as spoken language back then, and Ancient Greek, including the above-mentioned Atticised Greek, is beneficial for text classifiers detecting incorrect recognition output. Furthermore, by covering a period of more than seven centuries, the already challenging nature of recognition (Platanou et al., 2022) is increased, hence allowing a more diverse material for our investigation on post-correction. Errors in handwritten text recognition are often due to different hands (scribes) and scripts that evolved for many centuries. Printed text, on the other hand, exists for a much narrower period, there are no hands to cause similar errors,[1] and no clitics. As we show in this work, lower recognition error rate means more difficult classification of erroneous lines. By contrast, recognised handwritten text is often hard to parse even for humans, which makes an easier classification challenge and a much more difficult

---

[1]Fonts pose a similar problem, yet less challenging when compared to hands, due to their non-standardised form and high variation.

error-correction task, as shown in Table 1.

### 1.3 The contributions

We explore the extent to which classification to flawless/flawed texts can improve the post-correction of recognised output, by avoiding adversarial errors introduced when applied to flawless texts. We benchmark machine- and deep-learning-based text classifiers, showing that a Transformer that is pre-trained in modern Greek and then fine-tuned to detect erroneously recognised Byzantine Greek (10th-16th CE) text achieves an Average Precision of 95%. When the same Transformer is further pre-trained in ancient Greek prior to fine-tuning, the score increases to 97%. We quantify the benefits of employing our text classifier in the real world, by training an MT5 (Xue et al., 2021) encoder-decoder error-corrector, which altered 80% of the flawless texts. If our best-performing classifier had been applied prior to correction, the percentage would have been decreased to 2% with a trade-off of 2% texts being incorrectly classified as flawless.

The better performance of a Transformer that is not pre-trained only in modern Greek can be explained by the fact that Byzantine Greek combines language elements from both, ancient and modern Greek. Since our study covers a broad historical period, we separated the evaluation dataset based on the century in order to investigate this hypothesis. Our results show that the detection of erroneous recognition depends on the chronology of the manuscript and that further pre-training in ancient Greek benefits the classification of lines from manuscripts in the following three centuries: 10thCE, 11thCE, and 13thCE. By following a similar segmentation-based evaluation method for the error rate level, instead of the century, we show that higher error rates make an easier classification goal. By contrast, the problem is harder for low error rates, when flawed and flawless texts look alike.

## 2 Related work

Our work focuses on the text classification of recognised handwritten text. In the absence of published studies for handwritten text, however, we summarise work focused on the recognition of printed text. Printed and handwritten materials differ, because the latter often yields scrambled recognition output (Table 1). This is probably what motivated Ströbel et al. (2022) in using the perplexity of language models to detect the erroneous output in an unsupervised manner. On the other hand, although we consider that recognition errors are overall more for handwritten text compared to printed material, the quality of recognition can vary significantly for the former (Hodel et al., 2021), as is also shown in Fig. 1, and does not always come with a high error rate. Therefore, the detection of erroneously recognised handwritten text is also related to that for printed text, regarding their low error output.

**Erroneous recognised printed text detection**

Initial work using rule-based approaches could not address all errors, and was followed by machine and deep learning approaches (Lyu et al., 2021). Standard approaches include support vector machines on top of n-gram-based features (Dannélls and Virk, 2021; Virk et al., 2021). Jatowt et al. (2019) used a gradient tree boosting binary classifier on top of various features, including character and word n-grams, part-of-speech, token frequency based on automatically created resources.

Amrhein and Clematide (2018) performed detailed analyses and experiments of error detection and post-correction with various statistical and neural machine-translation methods. In general, they found that the former perform better in error correction, while the latter models are better in error detection. However, they concluded that there is not a single method that works best on all datasets while the results are highly affected by the data a model is trained on. Furthermore, they suggested that post-correction of recognised printed text should focus more on the improvement of error detection.

Nguyen et al. (2020) used BERT pre-trained on named-entity recognition for token classification in order to perform error detection. After subtoken tokenization they obtain Glove or Fasttext word embeddings, which are combined with segment and positional embeddings and given as input to BERT (Devlin et al., 2018). The hidden states are fed to a dense layer on top that classifies each token as erroneous or not. Although interesting, token-level detection is not suited for high-error settings.

**The ICDAR challenge**

ICDAR organised a competition focused on post-correcting text recognised from newspapers, shopping receipts, and other printed sources, including two subtasks: error detection, i.e., to detect the position and the length of the errors, and error correction (Chiron et al., 2017; Rigaud et al., 2019).

In 2017, the best-performing method for the error detection subtask employed probabilistic character error models and weighted finite-state transducers while in 2019, the best was a pre-trained multilingual BERT with convolutional and fully-connected layers on top that classified each sub-token as erroneous or not. Schaefer and Neudecker (2020) followed a similar classification scheme, but focused on characters and not sub-tokens. Then, by excluding lines with at least one classified character from post-correction, they showed that false alteration (i.e., of correctly OCRed lines) by an encoder-decoder model could be avoided. Our work shows that casting this problem as a text classification task can lead to high accuracy for handwritten text, although similar benefits may apply to printed text.[2]

## 3 Data

**The written language** of the manuscripts and papyri used in this study is Byzantine Greek.[3] Within these texts, morphological categories such as the optative, the pluperfect, and the perfect have disappeared, while others such as the dative case have gradually decreased. Infinitives and participles are still there in the texts, as reminiscents of the classical tradition, encouraging one to treat the language as a unique variant, different from modern Greek. There are several spelling conventions that deviate from the older orthographic rules while the ancient punctuation signs are still in use, albeit not always with the same function. Therefore, following Classical and preceding Modern Greek, this language can be considered a combination of the two.

**The transcriptions and the HTR output** were provided by the organisers of the recent HTREC challenge,[4] which regarded the automated correction of HTR errors in Byzantine Greek (Pavlopoulos et al., 2023). Selected images of handwritten text in respective manuscripts (10th to 16th CE) were transcribed by a human expert. The training dataset of the challenge consists of 1,875 lines of transcribed text by a human expert and by an HTR model (examples shown in Table 2). Transkribus (Kahle et al., 2017) was used by the organisers to produce the system transcriptions, trained on seven images one per century. Our investigation of the CER of the opted HTR model reveals

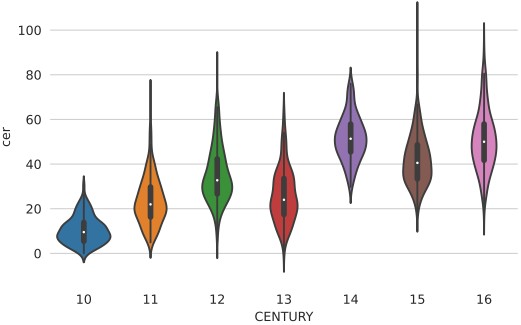

Figure 3: Violinplot of CER per century

that a diversity of errors is produced, from very few to many per line (Fig. 1). This fact makes this dataset optimal for our training purposes, because we need a diversity of errors to train our classifiers.[5] Furthermore, the errors are not uniformly distributed across centuries (Fig. 3), motivating a time/CER-based segmentation for evaluation purposes (discussed in Section 4.2).

**Our classification dataset** consists of the 1,875 human-transcribed texts (lines), treated as the flawless negative (0) class, and the respective system-transcribed texts, treated as the flawed, positive (1) class.[6] This resulted in total to an almost perfectly balanced dataset of 3,744 text lines, which we split to train, development and test subsets by using a 80/10/10 ratio. The average length of the transcription is 40.10 characters (6.83 words), and that of the HTR output is 37.09 characters (5.89 words).

## 4 Experimental analysis

We opted for machine and deep learning benchmarks, with different transfer learning settings.[7]

**Machine learning benchmarks**

Machine learning methods include a variant of Naive Bayes and Support Vector Machines (NB-SVM), which is often used as a baseline for text classification (Wang and Manning, 2012).[8] Also, we experimented with a Random Forest and a multilayer perceptron (MLP) using character n-grams

---

[2]The difference lies in using a `sigmoid` function (or so) on top of the text representation, instead of one per character.

[3]Also known as Medieval Greek or simply Byzantine.

[4]https://www.aicrowd.com/challenges/htrec-2022

[5]Fewer errors will harm class balance while similar errors will harm generalisation of the classifiers.

[6]Six lines were excluded from the positive class, because a perfect transcription was achieved.

[7]All experiments were performed using Google's Colaboratory, using a Tesla T4 GPU card. Our large language models comprise 113 million parameters.

[8]We used the implementation of KTRAIN with default parameters (Maiya, 2020).

| Transcription by expert | HTR output |
|---|---|
| ἐγγινομένα πάθη μὴ σβεννύντες ἀλλὰ τῇ εκλύσει (the born-in passions not extinguishing but the release) | ἐγγενομεναπαδημησμεννωτες ἀλλατῇε χλησει |
| τοῦ βίου τοῦ καθ ᾽ εαυτοὺς πολλὰ γίνεσθαι συγχωροῦν (of the life of themselves many happening forgive) | του β ου του καλεαυτοὺς πολλαγινεσθαι συγχωρ όν |
| τες ἐμπυρίζουσι τὸν ἀμπελῶνα ἀλλὰ καὶ ὁ διὰ τες (- set on fire the vineyard but and the due to the) | εμπυριζου σιμαμπελῶνα ἀλλακαι ὅδξα |

Table 2: Human (on the left) and HTR-based (on the right) transcription (lines) from the 'Commentary on Isaiah' by Basil the Great (fol. 75r, Oxford, Bodleian Library MS. Barocci 102)

($n \in 1, 5$), term-frequency inverse document frequency, and default parameters otherwise.[9]

**Deep learning benchmarks**

Deep learning methods comprise a recurrent neural network using Gated Recurrent Units (GRU) and trainable word embeddings of 300 dimensions, originally pre-trained on (modern) Greek (Joulin et al., 2017).[10] Also, we experimented with the state-of-the-art in text classification, which is BERT (Devlin et al., 2018) and its variants (Li et al., 2022; Minaee et al., 2021). In specific, we also fine-tuned two BERT models pre-trained in Greek. The first (GreekBERT) is pre-trained in modern Greek (Koutsikakis et al., 2020) while the second extends the first by being further pre-trained in ancient Greek (Singh et al., 2021).[11]

**Evaluation measures**

We opted for the $F1$ score per class, the Area under the Receiver Operating Characteristic Curve (AUC), and Average Precision (AP) that is preferred in class-imbalanced settings. All evaluation measures were implemented in SCIKIT-LEARN.[12]

### 4.1 Text classification benchmark

Table 3 shows the results of all our text classification benchmarks, including a random baseline (responding uniformly) that draws the lower performance limit. NBSVM is the worst machine learning model, followed by Forest and MLP. The latter achieves promising results, close to GRU (better in AP and $F1^-$). GRU, on the other hand, is clearly

|  | AP | AUC | $F1^+$ | $F1^-$ |
|---|---|---|---|---|
| Random | 0.52 | 0.50 | 0.49 | 0.47 |
| NBSVM | 0.66 | 0.65 | 0.60 | 0.51 |
| Forest | 0.64 | 0.65 | 0.64 | 0.50 |
| MLP | 0.79 | 0.79 | 0.73 | 0.69 |
| GRU | 0.79 | 0.79 | 0.68 | 0.71 |
| GreekBERT:M | 0.95 | 0.94 | 0.88 | 0.88 |
| GreekBERT:M+A | **0.97** | **0.97** | **0.90** | **0.91** |

Table 3: Evaluation with Average Precision (AP), AUC, $F1$ for the classification of HTRed texts to flawed (+) or flawless (-) transcription. Random (uniform) classification was used as a baseline. GreekBERT was pre-trained in modern (:M) or in modern and ancient Greek (:A).

outperformed in all metrics by GreekBERT. Two GreekBERT models were used, one pre-trained in modern Greek (:M) and one in modern and then in ancient Greek (:M+A). Byzantine Greek, as already discussed (Section 3), combines language elements found in both ancient and modern Greek. This is probably why GreekBERT:M+A, which was pre-trained in modern and then in ancient Greek, was better compared to GreekBERT:M, which was pre-trained only in modern Greek.[13]

### 4.2 Evaluation per century

We separated the lines of the test data based on the century of the manuscript that they came from. For each century, then, we evaluated the two Transformers, GreekBERT:M and GreekBERT:M+A. As can be seen in Fig. 4,[14] GreekBERT:M+A is better on the 10th (five percent units) and 11th century CE, with models performing similarly on the 12th and after the 13th century, when both models achieve a

---

[9] https://scikit-learn.org/stable/modules/generated/sklearn.neural_network

[10] https://fasttext.cc/docs/en/crawl-vectors.html

[11] Ancient Greek texts, which can be used for pre-training, are much fewer in number compared to publicly available modern Greek data. Hence, the option of pre-training directly in ancient Greek was not considered viable.

[12] https://scikit-learn.org/stable/

[13] We verified the consistency of this finding by repeating the experiments three times, reporting an average AUC of 0.97 over 0.94 (st.d. lower than 0.05).

[14] For statistical significance, we sampled twenty lines per century, repeating the sampling process ten times. We focus on Average Precision, but similar findings were observed with the other metrics.

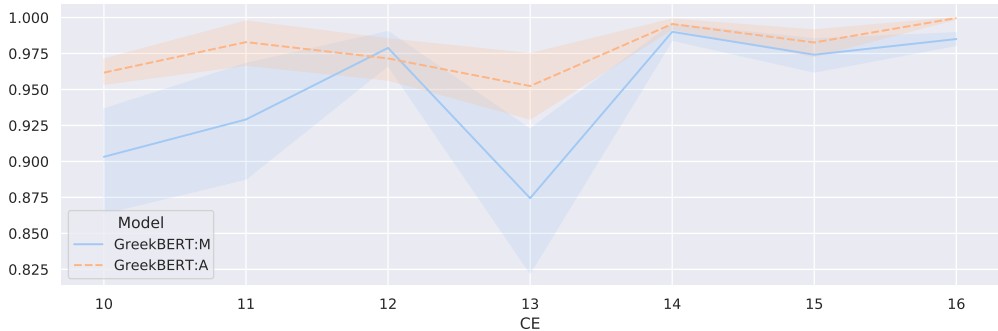

Figure 4: Average Precision per c. CE of GreekBERT:M, pretrained on modern Greek (solid), and GreekBERT:M+A, further pre-trained on ancient Greek (dashed) - error bands show the 95% bootstrap confidence intervals

higher score. On the 13th century, the performance of both models drops, but considerably more for GreekBERT:M. The 14th, 15th and 16th century were more challenging in HTR (Platanou et al., 2022), which could explain the better performance of the classifiers. That is, more errors in the recognised output could potentially help the classifier distinguish between the two classes (correct/not).[15] Based on the previous observations, we report three findings:

F1. *The classification performance depends on the chronology of the manuscript.*

F2. *Older manuscripts are better handled by further pre-training on ancient Greek.*

F3. *Lines from recent manuscripts pose an easier classification challenge to both GreekBERT models.*

### 4.3 Evaluation per error zone

Our dataset is balanced regarding whether the recognition output is erroneous (positive class) or not, yet it remains imbalanced regarding the CER level (see Fig. 1). In order to assess the two Transformers across different input types, we rolled a window of ten units, from low to high CER values, and we assessed the models per window (Fig. 5).[16]

**High-CER** lines (i.e., lines with CER of thirty or more) appear to set an easy classification task to both models. Performance increases as the CER increases. This is reasonable considering that lines with a high CER comprise many errors and are hard to parse even by humans. This characteristic, however, makes them easier to distinguish from flawless lines, which comprise no errors from the recognition.

**Low-CER** lines (i.e., lines with CER of zero to twenty) lead to a different picture, since Greek-BERT:M+A is consistently better, overall, than GreekBERT:M. Low CER means small differences between the two texts (flawless-flawed), which makes the classification task more difficult. In this zone, further pre-training on ancient Greek had a bigger impact, compared to zones with a higher error rate.

## 5 Discussion

In this section, first, we assess the possibly adversarial nature of post-correction. Then, we describe a synthetic error detection application, followed by a century-based error analysis we performed.

### 5.1 Substantiality assessment

Error-correction systems may introduce errors, even when applied to flawless HTRed texts. To assess the substantiality of this claim, we experimented with a neural encoder-decoder post-corrector. In specific, we employed a multilingual variant of the T5 Text-to-Text Transfer Transformer (Raffel et al., 2019), called MT5 (Xue et al., 2020), which was pre-trained on a dataset covering 101 languages.[17] MT5 has been used for Grammatical Error Correction in modern Greek and sets a strong baseline for HTR Error Correction (Korre and Pavlopoulos, 2022). We fine-tuned this model on the 1,875 training instances of the HTREC

---

[15]It remains unclear to the authors whether more hands were present during these centuries, explaining the performance of the HTR and the classification models, or if this was due to another reason; e.g., writing became more cursive over time.

[16]We sampled thirty lines, repeating sampling ten times.

[17]The Greek language was on the 20th position with 43 billion tokens extracted from 42 million pages

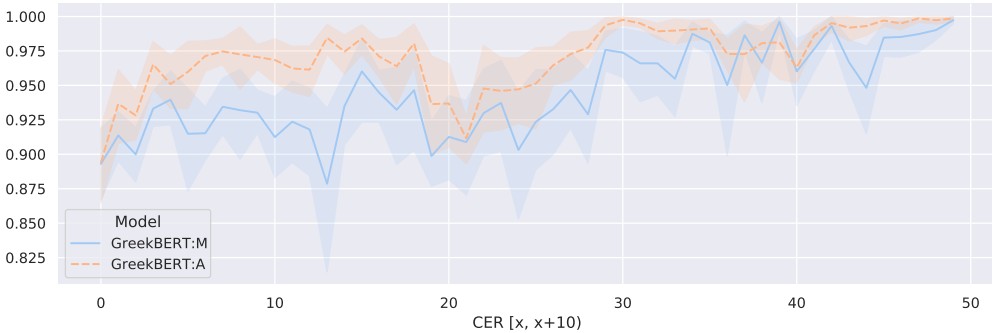

Figure 5: Average Precision of GreekBERT:M, pretrained on modern Greek (blue, solid), and GreekBERT:M+A, further pre-trained on ancient Greek (orange, dashed), using a rolling window over CER (left edge shown horizontally) - error bands depict 95% bootstrap confidence intervals

challenge, by encoding the system- and decoding the human-transcriptions.[18] This model is a neural error correction baseline that reduced the CER in 19 out of the 180 HTRed texts in the official HTREC2022 evaluation set. We applied this corrector on the respective 180 transcriptions of these data (the ground truth evaluation instances in HTREC2022) and found that 144 (80%) were mistakenly altered by our MT5 corrector. For example, the line 'αιν τους μηδους οιτινες ουτε αργυρτω παρα' was altered to 'αιν τους μηδους οι τινες ουτε αργυρτω παρα' (word division, 4th word). Similarly, 'χευσης μη φονευσης μη κλε' was altered to 'χλευσεως μη φονευσεως μη κλεπτ'. Most of these adversarial mistakes (all but the top three of Table 4) would have been avoided if our GreekBERT:M+A classifier had been applied prior to the application of this neural corrector. Conversely, only four (i.e., the rest in Table 4) would have been mistakenly found as correct from the system-transcribed instances.

## 5.2 A synthetic application of error detection

We experimented with the HTREC2022 evaluation dataset, which was used to assess the participating systems. This dataset comprises 180 lines that were transcribed from nine images. Out of these lines, one had zero CER achieved by the HTR model (perfect transcription), which was correctly classified by our GreekBERT:M+A model. We then performed two analyses, first by applying our model on the flawless human transcriptions (as if a transcription was the HTR output) and then by applying

it on the flawed system recognitions (the same lines were used). The human transcriptions were correctly classified as flawless, all except three (1.7% error rate), which are shown in the top three rows of Table 4. The system recognitions were also correctly classified as flawed, all except four (2.3% error rate), shown in the last four rows of Table 4.

## 5.3 Error analysis

An error detection by experts in Table 4,[19] showed that no clue (indicating an error) was found for the first line (except from the rare 'φιλακόλουθός' with double accent). As is shown in Figure 6 (a), which depicts the attention on the subwords of this line, this was indeed the network's focus.

On the second line, word 'ισμαηλήτι' was considered as the outcome of misspelling of the word 'ισμαηλιτη', matching the algorithm's decision. In this case, it was the ending of the word that attracted attention (Fig. 6), but the network focused more on the article 'των' preceding the word rather than on the word itself. The article is in plural form, not matching a word in singular. Apparently, 'ισμαηλήτι' is not a misspelling, but part of the word 'ισμαηλιτιχων' that was corrupted.

In the last four lines, that were classified as correct, several flaws were raised, indicating a failure of the algorithm. For example, word divisions, such as in the case of 'ωσχαι' instead of 'ως χαι'; added characters, such as in the case of 'εὐυχης' instead of 'εὐχης'; character replacements, such as in the case of 'ζχετείαν' instead of 'ιχετείαν'.

---

[18] We used the SimpleT5 implementation (https://github.com/Shivanandroy/simpleT5) with 100 epochs, early stopping (patience of 2 epochs), 15 tokens max. length, batch size of 4, and best loss found at the 12th epoch (2.64).

[19] No access was given to the manuscript or the human transcription, for fair comparison with the setting of the algorithm.

| TRANSCRIPTIONS AND HTR OUTPUT | GREEKBERT:M+A | GT |
|---|---|---|
| ἐγὼ δ ἀεί πως φιλακόλουθός εἰμι (I am always readily following) | F | N |
| λα και γεδεων εχ των σκυλων των ισμαηλητι (- and Gedeon from the dogs the -) | F | N |
| δραμων εις και γονυπετη (having run to and falling on the knee) | F | N |
| τιθεις τας χειρας επαυτα (having put the hands on them) | F | N |
| ζχετείαν ὠδέσποτα, καὶ ευμενῶς ἀκουσον τῶν τῆς προς εὐψχης (- - - and favourably listen to the of the prayer) | N | F |
| ἐγενόμην ἴς. αραή τήςαε μεό τιεγεζήθης ἐλγειεμοητιγριος (became - - - - - -) | N | F |
| ωσχαι της συνεχους διδασκαλιας τους βα (- of the continuous instruction the -) | N | F |

Table 4: HTRed lines and transcriptions that GreekBERT:M+A mistakenly predicted as flawed (F) or not (N) - the ground truth (GT) on the right shows whether there was an error or not

## Error patterns

Next, we proceed with an analysis of the recognition errors, using data from three centuries. We opt for 16th CE, the century on which both our models perform considerably well (Fig. 4). Next, we focus on the 14th CE, going back two centuries, but still observing good performance by both models. Last is shown the 13th CE, where GreekBERT:M performs much worse than GreekBERT:M+A.

**16th CE** This is the data group with the highest average CER (see Figure 3). What seems to be challenging in this data group is the fact that there is a variety of punctuation marks which are sometimes read by the model as letters because of their shape. An illustrative example of this case is the word 'πλοῦς' which is read by the model as 'πλουοο'. This is due to a comma following the word 'πλοῦς', which is very similar to the last letter of the word (ς). There are also many cases of ligatures in which the model fails to recognise both letters, especially the one written on the top of the ligature, such as the letter 'τ' in the combination 'τρ' as one can see in the following instance; 'ὢ πατρὶς' read as 'ως ὤπαρις'.

**14th CE** The data from the 14th century show the second highest CER on average (see Figure 3). This is partly because of a great number of ligatures appearing in this data group. A common ligature among this group is the one which represents the group of the letters 'ε' and 'ρ'. In this ligature it happens that the letter 'ε' is written above the letter 'ρ'. The way the model interprets this structure can be seen in the following case; 'φέρ᾽ ἐπ᾽ αὐτὴν ἴω', where 'φέρ' consists of one letter (φ) and one ligature (ερ) is read by the model as 'φρεπαυ την ιςω'. The model fails reading correctly the letter combination and returns only the letter 'φ' and one of the two letters involved in the 'ερ' combination (ρ), which is actually the one written in the position next to the letter 'φ'. The same instance shows clearly that there is a tendency towards adding characters because of different drawing representations of characters found in this data group. The letter ω can be considered such a case. In the previous example, we can see that the word 'ἴω' is read by the model as 'ιςω'. This is because, in this instance, the letter 'ω' has the shape of a laid eight. The same letter has the usual shape in other cases in the text of this century, which was quite confusing for the model and led the model to decide that it should be a combination of both 'σ' and 'ω', since the former is quite close to the half laid-eight shape.

**13th CE** The data group that is easiest to read for the model seems to be the 13th century one. In this group, errors appear mainly because of special letter shapes and abbreviations. Such a letter shape is that of the letter 'ε' which is written as the letter 'δ' with an added line pointing upwards. A characteristic example of the failure of the recognition task is this one; 'ἕως' is read as 'δως'. Here the model has clearly classified the 'ἕ' as a 'δ'. Regarding the abbreviations representing character groups, one can find among others the following

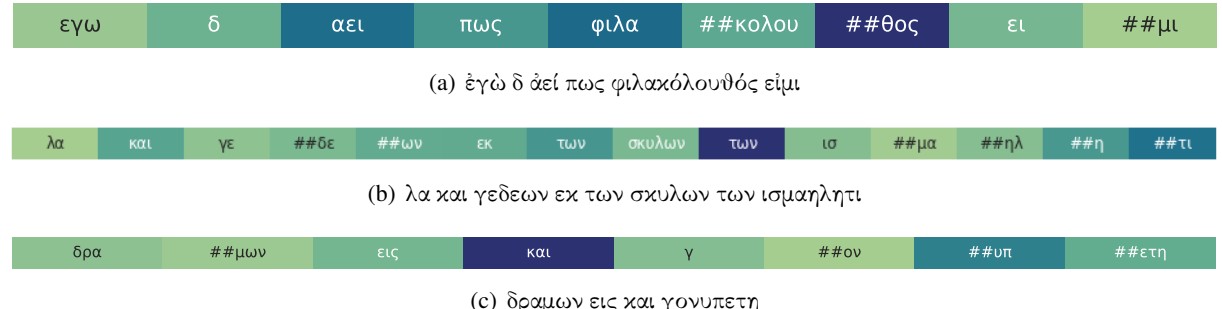

| εγω | δ | αει | πως | φιλα | ##κολου | ##θος | ει | ##μι |

(a) ἐγὼ δ ἀεί πως φιλακόλουθός εἰμι

| λα | και | γε | ##δε | ##ων | εκ | των | σκυλων | των | ισ | ##μα | ##ηλ | ##η | ##τι |

(b) λα και γεδεων εκ των σκυλων των ισμαηλητι

| δρα | ##μων | εις | και | γ | ##ον | ##υπ | ##ετη |

(c) δραμων εις και γονυπετη

Figure 6: The attention of GreekBERT:M+A (darker is more) for the three top (missclassified) lines of Table 4 - the symbol ## is used to concatenate a subword with the preceding one during tokenisation (δραμων is δρα ##μων)

one; 'αν'. In the case of 'οὐκ ἴσχυσαν', where the last two letters are represented by the abbreviation, the model reads 'οὐκισχω', where 'ω' has replaced the abbreviation because of similarity in shape.

## 5.4 Language and script generalisation

Our work can potentially inspire analogous pre-training in other languages with a long documented history. To provide an example: Old, Middle, and Modern English and Japanese (Lyovin, 1997, pp. 2) can qualify for pre-training on both Old and Modern data, as in the Greek language in our case. The scripts our results are based on, however, are very common both in Greek and Latin manuscripts while common handwritten character shapes, e.g., epsilon (E) (Chambers and Chambers, 1891, pp. 703), may illuminate error patterns in Latin scripts.

## 6 Conclusion

This work showed that detecting erroneous recognised lines improves the accuracy of handwritten Byzantine text recognition, aiming to assist the disciplines struggling with vast corpora and little man power. By contrast to printed text, errors in handwritten text vary greatly, some of which are easier and others difficult to catch. Our experiments showed that two BERT models, one pre-trained in modern and the other pre-trained in modern then in ancient Greek perform well for the task of erroneous handwritten recognised text classification. The latter performs better, especially for manuscripts from older centuries and for lines with higher recognition error rate. By using a neural text-to-text Transformer for error correction along with our best performing classifier, we found that adversarial behaviour is not uncommon in the neural corrector's output which could have been significantly limited if a BERT classifier, pre-trained and fine-tuned, had classified the input first.

**Directions of future work** comprise the development of a larger dataset, which, although not easily developed in our domain of focus, it could potentially allow us to draw more conclusions and lead to further findings. Furthermore, we plan to extend our dataset with calibrated evaluation subsets, in order to gain more perspective on the performance of systems for specific error types, including ones that occur naturally and are not meant to be corrected.

## Limitations

- Our study is focused on Byzantine Greek, from the 10th to the 16th century. Our hypothesis, that text classification could assist the post correction of HTRed output, should be studied for more languages and covering a longer period to draw general conclusions.

- By contrast to OCR, where low CER values are reported, HTRed output can significantly differ from the human transcription. For example, correct text may be misplaced (e.g., due to mistaken word division) or by the introduction of an extra character (e.g., the last word of the fifth line in Table 4). Hence, the creation of ground truth at the character level is very challenging in this context, hindering experimentation at the token or the character level, as in OCR (Schulz and Kuhn, 2017; Schaefer and Neudecker, 2020).

- Sensitivity analysis has not been performed. Although we repeated our experiments to verify the validity of our findings regarding GreekBERT, we have not experimented with cross validation over the centuries, as in (Platanou et al., 2022). Training, however, an algorithm on texts from specific centuries, then as-

sessing on ones from previous or future ones, could potentially reveal interesting patterns.

- Diplomatic transcriptions may lead to naturally-occurring errors (i.e., ones the editors would want to keep, to stay as close to the original as possible), which our proposed classifiers could and probably would classify as incorrect recognition output. This limitation could be bypassed if the proposed systems are used complementary with an expert, not trying to substitute one. For example, the expert could view only lines the classifier labelled as incorrect, bypassing or acknowledging errors which were meant to be there.

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
