# OpenReview forum: "Detecting Erroneously Recognized Handwritten Byzantine Text"
_EMNLP/2023/Conference — EMNLP 2023 Findings_

### Official Review · Reviewer_nm3Z · 2023-07-19

**Soundness:** 3

**Excitement:**

3: Ambivalent: It has merits (e.g., it reports state-of-the-art results, the idea is nice), but there are key weaknesses (e.g., it describes incremental work), and it can significantly benefit from another round of revision. However, I won't object to accepting it if my co-reviewers champion it.

**Missing References:**

161: Hodel et al. https://openhumanitiesdata.metajnl.com/articles/10.5334/johd.46

166: maybe include Springmann et al. https://arxiv.org/pdf/1606.05157.pdf






**Paper Topic And Main Contributions:**

This paper evaluates the use of classifiers to detect erroneously recognised text from HTR models that have been applied to Byzantine text. The authors show that Transformer-based classifiers exceed in this task and can be a great help to evaluate automatically recognised texts. As such, the paper makes an important contribution to the field of the Digital Humanities, and especially in the realm of digitisation procedures. One of the most interesting findings is that the performance of classifiers differs based on the century a text had been written in. This shows that the LMs that are behind the classifier (here BERT) are sensitive to language change, or rather, it implies that while training HTR systems, data from several periods should be included.

**Questions For The Authors:**

Question A: You say early on that you dover 7 centuries. How does Byzantine Greek change over this time? Shouldn't you include 2 or 3 sentences around line 097? If it changes, how do you think these changes affect the performance of models? --> after having read on, you elaborate on this in Section 3. However, the question about the impact these changes had remains.

Question B: For editions of texts, editors usually want to stay as close to the original as possible. That is, during writing, errors naturally occur. How would you make sure that automatic correction does not touch such words? And, more importantly for your work: a classifier would probably label such a word as error, when indeed the model provided the "correct" transcription. How would you treat such cases? --> again, after having read on, you hint at this in Section 5.1. Still, a qualitative analysis of the errors introduced is missing. I think this would be something your contribution would profit from!

Question C: How much do the hands differ in the dataset? Could the variance of performance of the HTR system be due to the greater variance of the hands?








**Reasons To Accept:**

The main reasons to accept the paper are:
- the careful analysis of the dataset with the help of the results the authors attained
- the analysis of error patterns
- the discovery of time-period-based performance differences
- the fact alone that work on Byzantine is presented, which is not so frequent

I enjoyed the read and found it interesting.

**Reasons To Reject:**

The paper has some flaws:
- I'm missing the read thread through the paper. I could not always follow the arguments without re-reading and some claims need to be substantiated! (see comments)
- We could attest the paper some kind of lack-of-novelty, but the usefulness of the approach and the evaluation in a new (the Byzantine) setting make up for this.
- I would have expected some more resources which have tackled the problem in the past.

**Reproducibility:**

5: Could easily reproduce the results.

**Reviewer Confidence:**

5: Positive that my evaluation is correct. I read the paper very carefully and I am very familiar with related work.

**Typos Grammar Style And Presentation Improvements:**

I include general questions here. I do not expect a detailed answer for them, but are thought as food for thought for the authors. Some comments are high-level and I leave it to the authors if they want to consider them. Some are just common thoughts.

015: precision with respect to detecting incorrect output?

029: computerised --> I'd prefer digitised

030: HTR to an

033: ideally yes, but this not always the case!

036: delete :

038: delete ;

040: from the

043: delete ,

040-046: I would like to have some backup for this claim (it is true, but maybe you mention some refernces where this is actually the case).

Figure 2: should maybe be Figure 1. Moreover, change \citep to \cite in caption. Also, what is the y-axis? # of docs?

FIgure 1: maybe use an image with higher resolution?

071: delete ,

075: introducing instead of mitigating  errors that ...

078: delete e.g.,

079: rather choose characters instead of letters and delete ,

082-085: that is a strong claim. any backup? what about long distance relationships? what about attention-based models that take huge contexts into account?

085-091: maybe you introduce an example for this.

104: no clitics. and: there might not be hands, but different fonts

161: here you say HTR does not always come with high error rates, while in 030 you say in can lead to high error rates and in 054 you say it leads to more errors. this is somewhat inconsistent. Maybe cite Hodel et al. (see missing references) for an indicator for what is possible for HTR and what is regarded as good, mediocre or bad in terms of CER.

159: as far as I know, Ströbel et al. evaluated on HTR and not printed texts

214: instead of

footnote 1: or so --> not very scientific

290: More specifically,

Table 3: call it Random Forest, same in 307

311: pre-trained on -> several instances

Figure 4: name of y-axis missing, same for Figure 5

348: thirty or more what, errors per line?

352: for humans

356: zero to twenty what?

365: section we first assess ...

459: characters

in general: speak of characters instead of letters

481: reading the character combination correctly

---

> ### Author Rebuttal · Authors · 2023-08-28
>
> We highly appreciate the constructive feedback and thoughtful suggestions provided by the reviewer. We commit to revise our work accordingly. Please read our thoughts on the raised concerns and how we addressed them below:
>
> > Question A: You say early on that you dover 7 centuries. How does Byzantine Greek change over this time? Shouldn't you include 2 or 3 sentences around line 097? If it changes, how do you think these changes affect the performance of models? --> after having read on, you elaborate on this in Section 3. However, the question about the impact these changes had remains.
>
> Correct. We will add the following in line 096:
>
> “The development of the Byzantine Greek language entails a comprehensive examination of written language throughout the centuries under study. In essence, Byzantine Greek emerged as an academic style known as Atticised Greek, incorporating religious topics. As time progressed, elements of contemporary spoken Greek gradually became integrated into the language [1]. As the results of our study also show, the combination of pre-trained data on Modern Greek, resembling what was considered as spoken language back then, and Ancient Greek, including the above-mentioned Atticised Greek, is beneficial for text classifiers detecting incorrect recognition output.”
>
> [1] Horrocks, Geoffrey. 2010. "Greek: A History of the Language and its Speakers". Second Edition. Malden, MA: Wiley-Blackwell., pp.220-221
>
> > Question B: For editions of texts, editors usually want to stay as close to the original as possible. That is, during writing, errors naturally occur. How would you make sure that automatic correction does not touch such words? And, more importantly for your work: a classifier would probably label such a word as error, when indeed the model provided the "correct" transcription. How would you treat such cases? --> again, after having read on, you hint at this in Section 5.1. Still, a qualitative analysis of the errors introduced is missing. I think this would be something your contribution would profit from!
>
> Correct. Diplomatic transcriptions may lead indeed to such errors, which our proposed classifiers could and probably would classify as incorrect recognition output. This is exactly why the proposed systems should be used complementary with an expert and not trying to substitute them. For example, the expert could view only lines the classifier labelled as incorrect, bypassing or acknowledging errors which were meant to be there. This is a limitation which we will include in our manuscript.
>
> > Question C: How much do the hands differ in the dataset? Could the variance of performance of the HTR system be due to the greater variance of the hands?
>
> The 14th, 15th and 16th century were more challenging in HTR [2], which might explain the better performance of the classifiers (Figure 4). That is, more errors in the recognised output could potentially help the classifier distinguish between the two classes (correct/not). Regarding whether more hands were present during these centuries, however, explaining the performance of the HTR and classification models, remains unclear to us and a material of future investigation. For example, another possible explanation is that writing became more cursive over time. We will clarify this.
>
> [2] Paraskevi Platanou, John Pavlopoulos, and Georgios Papaioannou. 2022. Handwritten Paleographic Greek Text Recognition: A Century-Based Approach. In Proceedings of the Thirteenth Language Resources and Evaluation Conference, pages 6585–6589, Marseille, France. European Language Resources Association.
>
>
> ### Missing references
>
> We added Hodel et al. and we will also add Springmann et al.
>
>
> ### Typos Grammar Style And Presentation Improvements
>
> We are grateful for the analytical feedback and we will fix all the points suggested. Please find below our response for specific comments:
>
> > 033: ideally yes, but this not always the case!
>
> We will add the word “typically”.
>
> > 040-046: I would like to have some backup for this claim (it is true, but maybe you mention some refernces where this is actually the case).
>
> We will add the following citation:
>
> [3] McGillivray, Barbara, Thierry Poibeau, and Pablo Ruiz. "Digital humanities and natural language processing:“Je t’aime... Moi non plus”." Digital Humanities Quarterly 14, no. 2 (2020).
>
> > Figure 2: should maybe be Figure 1. Moreover, change \citep to \cite in caption. Also, what is the y-axis? # of docs?
>
> The vertical axis indeed shows the number of lines (we will clarify) and we will consider moving our claim that “HTR output does not always come with a high error rate” up to the introduction along with the figure.
>
> > 082-085: that is a strong claim. any backup? what about long distance relationships? what about attention-based models that take huge contexts into account?
>
> This argument regards rule-based systems, lacking attentional or more generally neural components. It was used explicitly in order to provide an example regarding how errors may be introduced during post-correction. We will clarify this to avoid confusion.
>
> > 085-091: maybe you introduce an example for this.
>
> We will add the following example:
>
> "The line “λεντιον διεζωσεν εαυτόν” was HTRed as “λεντιον διεζωσενεαυτόν”. A neural corrector, then, which received as input the HTRed output, yielded “λεντιον διεζωσεν εαυτoν” (Pavlopoulos et al., 2023), correctly separating the two last words but incorrectly removing the accent."
>
> > 104: no clitics. and: there might not be hands, but different fonts
>
> Correct, we will add the lack of clitics. Also, we will add a footnote clarifying that fonts pose a similar problem yet less challenging when compared to hands, due to their non-standardised form and high variation.
>
> > 161: here you say HTR does not always come with high error rates, while in 030 you say in can lead to high error rates and in 054 you say it leads to more errors. this is somewhat inconsistent. Maybe cite Hodel et al. (see missing references) for an indicator for what is possible for HTR and what is regarded as good, mediocre or bad in terms of CER.
>
> We consider that recognition errors are overall more for handwritten text compared to printed materials. Also, Figure 2 shows that there is a greater error variety across lines, yielding lines of varying quality, from good to bad according to Hodel et al (2021). We will add the missing reference and consider rewording to avoid confusion.
>
> > 159: as far as I know, Ströbel et al. evaluated on HTR and not printed texts
>
> Indeed, Ströbel et al. worked on HTR. This is why we mention their work, after commenting in the previous sentence (lines 156-158) that handwritten material often yields scrambled recognition output. However, their work concerns the study of metrics for evaluating and ranking HTR models, which is different from the focus of our work and the related work we discuss.
>
> > 348: thirty or more what, errors per line?
>
> Lines with CER of 30+ (we will clarify)
>
> > 356: zero to twenty what?
>
> Lines with CER of 20- (we will clarify)

---

### Official Review · Reviewer_PgK8 · 2023-08-04

**Soundness:** 3

**Excitement:**

3: Ambivalent: It has merits (e.g., it reports state-of-the-art results, the idea is nice), but there are key weaknesses (e.g., it describes incremental work), and it can significantly benefit from another round of revision. However, I won't object to accepting it if my co-reviewers champion it.

**Paper Topic And Main Contributions:**

The authors provide a study of post-correction of Handwritten Text Recognition (HTR ) errors, with a focus on Byzantine Greek from the 10th to the 16th century. They also call out the findings related to classification task of detecting erroneously recognized lines. They use two BERT models to shore how pre-training on just modern Greek vs modern + ancient Greek differ in performance of classification task.

**Reasons To Accept:**

- Paper is well written and technically sound.
- Simple yet direct application & improvement in an area where lot of corpora is present but very less humans to look into.
- Looking through the lens of temporal aspect (proposed century-based approach) is very novel.
- The paper proposal could have some implications for other languages and scripts which exhibit similar patterns of variations over time.


**Reasons To Reject:**

- Some form of explicit ablation study might help create more perspective to view and solve this problem.
- Section on point-of-view about generalizability of the findings to other languages or scripts might help.
- Details around error type can also shed light on generalizability and may also help set new standards for post-correction process.




**Reproducibility:**

4: Could mostly reproduce the results, but there may be some variation because of sample variance or minor variations in their interpretation of the protocol or method.

**Reviewer Confidence:**

4: Quite sure. I tried to check the important points carefully. It's unlikely, though conceivable, that I missed something that should affect my ratings.

---

> ### Author Rebuttal · Authors · 2023-08-28
>
> We thank the reviewer for the feedback and the suggestions. Please read our response per point below:
>
> > Some form of explicit ablation study might help create more perspective to view and solve this problem.
>
> In future work, we plan to extend our dataset with calibrated evaluation subsets, in order to gain more perspective on the performance of systems for specific error types. We could not think of another ablation toward this goal but we could elaborate more if we misread this comment.
>
> > Section on point-of-view about generalizability of the findings to other languages or scripts might help.
>
> Correct. We mentioned this as a limitation of our work, but we will add the following paragraph in Section 5 to further address this point:
> *Our work can potentially inspire analogous pre-training in other languages with a long documented history. To provide an example, Old/Middle/Modern English and Japanese [1] can qualify for pre-training on Old-Modern data, as the Greek language in our case does. The scripts our results are based on, however, are very common both in Greek and Latin manuscripts while common handwritten character shapes, e.g., epsilon (E) [2], may illuminate error patterns in Latin scripts.*
>
> [1] Lyovin, A.V., Kessler, B. and Leben, W.R. (2017). "An introduction to the languages of the world". Oxford: Oxford university press., pp.2
>
> [2] "Chamber’s Encyclopaedia; A dictionary of universal knowledge". London: W. & R. Chambers., 1894, pp.703
>
> > Details around error type can also shed light on generalizability and may also help set new standards for post-correction process.
>
> Our work provided an extensive error analysis (Section 5.3), which included error patterns (please see also the 2nd reason to accept by `Reviewer nm3Z`). In future work, however, we plan to study more error types, such as the ones that occur naturally and are not meant to be corrected. We will clarify this.

---

### Official Review · Reviewer_ubKi · 2023-08-05

**Soundness:** 4

**Excitement:**

4: Strong: This paper deepens the understanding of some phenomenon or lowers the barriers to an existing research direction.

**Paper Topic And Main Contributions:**

The paper provides an approach for post-correction of the outputs from OCR in the Byzantine language. They showed that finetuning a masked language model on this language enhances the performance by 2%.

**Reasons To Accept:**

The paper provides a method for enhancing the output of the OCR by 2%. The paper provides an in-depth analysis of the errors.

**Reasons To Reject:**

- R1. The number of samples in the classification datasets is small ~2k samples, which is hard to determine the significance of the approach.
- R2. The approach is not compared with any baseline methods in the literature.

**Reproducibility:**

3: Could reproduce the results with some difficulty. The settings of parameters are underspecified or subjectively determined; the training/evaluation data are not widely available.

**Reviewer Confidence:**

2: Willing to defend my evaluation, but it is fairly likely that I missed some details, didn't understand some central points, or can't be sure about the novelty of the work.

---

> ### Author Rebuttal · Authors · 2023-08-28
>
> We thank the reviewer for the invested time and feedback. Please read our response below:
>
> > R1. The number of samples in the classification datasets is small ~2k samples, which is hard to determine the significance of the approach.
>
> In text classification settings, this size is not prohibitively small (see for example [1]) while we would like to note that larger datasets are not easily developed in our domain of focus (please see the 2nd reason to accept by `Reviewer PgK8`). We acknowledge that a larger dataset could potentially allow us to draw more conclusions and lead to further findings, hence we will propose it as future work.
>
> [1] Ravi Shekhar, Mladen Karan, and Matthew Purver. 2022. CoRAL: a Context-aware Croatian Abusive Language Dataset. In Findings of the Association for Computational Linguistics: AACL-IJCNLP 2022, pages 217–225.
>
> > R2. The approach is not compared with any baseline methods in the literature.
>
> This remark is not entirely clear to us. Besides the two Transformers and the machine learning algorithms we trained and assessed, we aren’t aware of another suitable baseline nor could we find one in literature. If the reviewer could provide more feedback, we could elaborate more.

---

### Meta-Review · Area_Chair_ZaVP · 2023-09-18

**Recommendation:** 4

**Metareview:**

The reviewers agreed that the paper tackles an under-resourced problem by means of thorough experiments and detailed dataset and error analyses. Criticism was centered around the generalizability of results, the small size of the dataset and a lack of novelty, which is made up for by the usefulness of the approach and the new Byzantine setting.

---

### Decision · Program_Chairs · 2023-10-07

**Decision:**

Accept-Findings

**Comment:**

The reviewers agreed that the paper tackles an under-resourced problem by means of thorough experiments and detailed dataset and error analyses. Criticism was centered around the generalizability of results, the small size of the dataset and a lack of novelty, which is made up for by the usefulness of the approach and the new Byzantine setting.